# Functional Studies of Deafness-Associated Pendrin and Prestin Variants

**DOI:** 10.3390/ijms25052759

**Published:** 2024-02-27

**Authors:** Satoe Takahashi, Takashi Kojima, Koichiro Wasano, Kazuaki Homma

**Affiliations:** 1Department of Otolaryngology—Head and Neck Surgery, Feinberg School of Medicine, Northwestern University, Chicago, IL 60611, USA; 2Department of Otolaryngology, Head and Neck Surgery, National Hospital Organization Tochigi Medical Center, Tochigi 320-0057, Japan; 3Department of Otolaryngology, Head and Neck Surgery, Tokai University School of Medicine, Isehara 259-1193, Japan; 4The Hugh Knowles Center for Clinical and Basic Science in Hearing and Its Disorders, Northwestern University, Evanston, IL 60208, USA

**Keywords:** pendrin, SLC26A4, prestin, SLC26A5, pendred syndrome, DFNB4, DFNB61, hereditary hearing loss, nonlinear capacitance

## Abstract

Pendrin and prestin are evolutionary-conserved membrane proteins that are essential for normal hearing. Dysfunction of these proteins results in hearing loss in humans, and numerous deafness-associated pendrin and prestin variants have been identified in patients. However, the pathogenic impacts of many of these variants are ambiguous. Here, we report results from our ongoing efforts to experimentally characterize pendrin and prestin variants using in vitro functional assays. With previously established fluorometric anion transport assays, we determined that many of the pendrin variants identified on transmembrane (TM) 10, which contains the essential anion binding site, and on the neighboring TM9 within the core domain resulted in impaired anion transport activity. We also determined the range of functional impairment in three deafness-associated prestin variants by measuring nonlinear capacitance (NLC), a proxy for motor function. Using the results from our functional analyses, we also evaluated the performance of AlphaMissense (AM), a computational tool for predicting the pathogenicity of missense variants. AM prediction scores correlated well with our experimental results; however, some variants were misclassified, underscoring the necessity of experimentally assessing the effects of variants. Together, our experimental efforts provide invaluable information regarding the pathogenicity of deafness-associated pendrin and prestin variants.

## 1. Introduction

The SLC26 family of membrane proteins includes two members, pendrin (SLC26A4) and prestin (SLC26A5), that are essential for normal hearing. Pendrin is an anion transporter required for normal development and maintenance of ion homeostasis in the inner ear [1,2]. Prestin is a voltage-dependent motor and mediates voltage-dependent somatic motility (electromotility) of cochlear outer hair cells (OHCs), which is essential for the high sensitivity and frequency selectivity of mammalian hearing [3,4,5,6]. Dysfunction or loss of these SLC26 proteins results in hearing loss. Pendrin-coding *SLC26A4* is a causative gene for Pendred syndrome (PDS, OMIM 274600), which is one of the most common types of syndromic hearing loss with enlarged vestibular aqueduct (EVA) and goiter, as well as nonsyndromic hearing loss (DNFB4, OMIM600791) [7,8]. Prestin-coding *SLC26A5* is associated with nonsyndromic hearing loss (DNFB61, OMIM613865). Since the advent of next-generation sequencing, a large number of deafness-associated variants have been identified in patients. As of January 2024, 704 and 16 variants have been identified in *SLC26A4* and *SLC26A5*, respectively (The Human Gene Mutation Database [9]). However, experimental assessments of the functional phenotypes of these genetic variants have been lagging behind in determining their pathogenicity with confidence. Previously, we quantitated the anion transport functions of disease-associated pendrin variants in order to understand the pathology underlying the various severity of PDS and DFNB4 [10]. Similarly, we characterized the functional phenotypes of prestin variants both in vitro and in vivo to determine their pathogenicity [11,12]. Those experimental efforts are indispensable to providing information regarding the effects of variants on protein functions, which in turn improves the consistency and accuracy in genetic diagnoses of patients. This is in line with the American College of Medical Genetics and Genomics (ACMG) and the Association for Molecular Pathology (AMP) guidelines for the interpretation of sequence variants adapted for genetic hearing loss [13].

In this study, we continue to assess the pathogenicity of deafness-associated pendrin and prestin variants in vitro to determine their roles in hearing loss in patients. For pendrin, highly reproducible fluorometric antiport assays previously developed by our group were used to interrogate the functional impacts of pendrin variants on the anion transport function [10], with a major focus on transmembrane (TM) 9 and 10 located within the core domain that accommodates an anion substrate and undergoes elevator-like motions with respect to the gate domain during the transport cycle. Since the number of variants identified was clustered around this functionally important region, we reasoned that the likelihood of identifying the pathogenic variants would be high. For prestin, nonlinear capacitance (NLC) measurement was used to determine the functional impacts of a few prestin variants reported to date on the motile function [14,15]. In addition, we compared our experimental results with predictions from AlphaMissense (AM) [16], a recently developed in silico tool that predicts variant effects using artificial intelligence. We found a decent correlation between AM pathogenicity scores and the antiport activity of pendrin variants, indicating good overall performance of this prediction tool (r = −0.6497). However, some of the variants were miscategorized, indicating that prediction tools still have limitations and that experimental efforts remain important to assess the pathogenicity of deafness-associated variants.

## 2. Results

### 2.1. Functional Characterization of Pendrin Variants

A recent cryo-EM study on mouse pendrin [17] revealed a homodimeric architecture that is similar to prestin [18,19,20,21] and other SLC26 proteins [22,23,24]. The transmembrane domain consists of the gate and the core domains, and the ion translocation pathway is located in between them (Figure 1A). The positive helix dipoles of TM3 and TM10 (their N-termini), together with a conserved basic residue (Arg^409^ and Arg^399^ in pendrin and prestin, respectively) in TM10, constitute a positively charged pocket that holds a variety of anion substrates. The N-terminus of TM10 is connected to TM9, which is located on the opposite side of the core domain. The importance of the conserved residues in TM9, TM10, and the linker portion connecting these two helices (Figure 1B) is suggested by the presence of multiple clustered missense variants identified in these regions: p.D380N (c.1138G>A), p.N382K (c.1146C>G), p.Q383E (c.1147C>G), p.Q383P (c.1148A>C), p.E384Q (c.1150G>C), p.E384G (c.1151A>G), p.I386V (c.1156A>G), p.A387V (c.1160C>T), p.G389R (c.1165G>A), p.G389R (c.1165G>C), p.S391R (c.1173C>A), p.S391N (c.1172G>A), p.N392S (c.1175A>G), p.N392Y (c.1174A>T), p.G396E (c.1187G>A), p.S399P (c.1195T>C), p.V402L (c.1204G>T), p.V402M (c.1204G>A), p.T404N (c.1211C>A), p.T404I (c.1211C>T), p.A406T (c.1216G>A), p.R409C (c.1225C>T), p.R409H (c.1226G>A), p.R409L (c.1226G>T), p.R409P (c.1226G>C), p.T410K (c.1229C>A), p.T410M (c.1229C>T), p.A411P (c.1231G>C), p.A411T (c.1231G>A), p.V412I (c.1234G>A), p.Q413R (c.1238A>G), p.Q413P (c.1238A>C), p.E414K (c.1240G>A), p.S415R (c.1245C>A), p.S415G (c.1243A>G), p.T416P (c.1246A>C) [10,25,26,27,28,29,30,31,32,33,34,35,36,37,38,39,40,41,42,43,44,45,46,47,48,49,50,51,52,53,54]. Among them, p.E384G, p.N392Y, p.S399P, p.R409H, p.T410K, p.T410M, p.Q413R, p.Q413P, and p.T416P were functionally characterized and found to significantly impair the anion transport function of pendrin in previous studies by us and others [10,42,55,56,57,58,59,60,61]. The present study extended our experimental efforts to define the functional consequences of missense variants identified in the TM9-TM10 region. To this end, we established HEK293T stable cell lines expressing pendrin variants in a doxycycline (Dox) dosage-dependent manner and conducted HCO_3_^−^/Cl^−^ and I^−^/Cl^−^ antiport assays alongside wild-type (WT) control as described in detail in our previous report [10]. Figure 2A,B show the results of 25 pendrin missense variants evaluated here for their Dox dosage-dependent HCO_3_^−^/Cl^−^ and I^−^/Cl^−^ antiport activities. Many of the variants exhibited vastly reduced or no activity, with only a few having WT-like activity (summarized in Table 1). These observations further affirm the importance of the conserved residues in TM9-TM10 in the core domain for the anion transport function of pendrin.

### 2.2. Functional Characterization of Prestin Variants

Mammalian prestin is unique among the SLC26 family of proteins as it functions as a voltage-dependent motor instead of a transporter. Prestin also shares the same overall similarities in structure as pendrin and other SLC26 proteins, implying a common molecular principle underlying transport and motor functions. Thus, it is conceivable that conserved residues between pendrin and prestin are of similar functional importance in these structurally similar proteins. Interestingly, however, the number of prestin missense variants reported to date is only 13, six of which are associated with autism with unknown pathophysiological relevance to hearing [54,62,63,64,65,66,67,68]. In this study, we characterized p.A100T (c.298G>A), p.P119S (c.355C>T), and p.S441L (c.1322C>T) prestin variants that are all associated with hearing loss. The locations of these three prestin missense variants are indicated in Figure 3A. We established HEK293T cell lines expressing these prestin variants and measured the NLC as a proxy for their motor function. Figure 4A shows the NLC of p.A100T and p.P119S prestin compared to WT. Although reduced, the NLC was detected for both p.A100T and p.P119S prestin, indicating some residual activity for these variants (Figure 4B). These Ala and Pro residues are relatively well conserved among the SLC26 proteins (Ala^104^ and Pro^123^, respectively, in pendrin, Figure 3B), and previous studies by us and others have shown that the p.P123S variant significantly impairs the anion transport function of pendrin [10,57]. In this study, we also measured the HCO_3_^−^/Cl^−^ antiporter activity for p.A104V pendrin (deafness-associated) and p.A104T pendrin (equivalent to p.A100T prestin but not identified in patients as of January 2024) and found that these two pendrin missense variants also significantly impair the anion transport function (Figure 5, left), suggesting the common importance of these conserved residues for pendrin and prestin.

Unlike p.A100T and p.P119S, the NLC of p.S441L prestin was undetectable, indicating that p.S441L completely abrogates the motor function of prestin (Figure 4A,B). Ser^441^ is located at the interface between the gate and core domains (Figure 3A, right). Equivalent residues at this site in the other SLC26 family members are either Ala or Gly (Ala^451^ in pendrin). These small residues may be important not to hinder the relative elevator-like movements of the core domain with respect to the gate domain. If true, p.S441L may sterically interfere with these interdomain motions. In line with this speculation, p.A451L in pendrin (equivalent to p.S441L in prestin) abolished the anion transport activity, whereas p.A451S and p.A451G did not (Figure 5, right). We also found that p.S441A prestin has a WT-like NLC (Figure 4A,B). These observations suggest the common importance of having a small residue at the Ser^441^ site in prestin and equivalent sites in the other SLC26 proteins.

**Table 1 ijms-25-02759-t001:** Summaries of the antiport rates of pendrin variants determined in this study. Numerical data from HCO_3_^−^/Cl^−^ and I^−^/Cl^−^ antiport assays and for HCO_3_^−^/Cl^−^ antiport assays are listed as indicated. *P*-values from one-way ANOVA with Fisher’s LSD tests compared to uninduced basal rates were listed below the mean ± SD values for each Dox concentrations. The numbers in the parentheses indicate the sample size. Comparison to WT was performed using the F-test on slope values as in our previous study [10]. Box with gray shades indicate statistically not significant (*p* ≥ 0.05). Asterisks (*) indicate variants not found in human patients (as of January 2024).

			Transport Activity [Mean ± SD (n)]					Dox-Dependence	
			Dox: 0.1 µg/mL	Dox: 0.3 µg/mL	Dox: 1 µg/mL	Dox: 3 µg/mL	Dox: 10 µg/mL	Transport Activity/ log_10_ [Dox] (Slope ± SE)	Comparison to WT (F-Test)
	**WT**	HCO_3_^–^/Cl^–^	0.597 ± 0.109 (10)	0.795 ± 0.125 (10)	1.102 ± 0.163 (10)	1.485 ± 0.290 (10)	1.731 ± 0.253 (10)	0.591 ± 0.040	not applicable
		[nM/sec]	*p* < 0.0001	*p* < 0.0001	*p* < 0.0001	*p* < 0.0001	*p* < 0.0001	*p* < 0.0001	
		I^–^/Cl^–^	0.08 ± 0.04 (3)	0.14 ± 0.01 (3)	0.18 ± 0.02 (3)	0.23 ± 0.04 (3)	0.21 ± 0.02 (3)	0.069 ± 0.012	not applicable
		[mM/sec]	*p* = 0.049	*p* = 0.0002	*p* < 0.0001	*p* < 0.0001	*p* < 0.0001	*p* < 0.0001	
** Figure 2 **	**p.Gln383Glu** (c.1147C>G)	HCO_3_^–^/Cl^–^	0.061 ± 0.023 (4)	0.072 ± 0.029 (4)	0.084 ± 0.060 (4)	0.125 ± 0.059 (3)	0.073 ± 0.020 (4)	0.015 ± 0.014	Function impaired
		[nM/sec]	*p* = 0.4446	*p* = 0.6253	*p* = 0.8678	*p* = 0.3807	*p* = 0.6457	*p* = 0.0034	*p* < 0.0001
		I^–^/Cl^–^	0.01 ± 0.003 (3)	0.012 ± 0.003 (3)	0.011 ± 0.004 (3)	0.015 ± 0.001 (3)	0.017 ± 0.005 (3)	0.003 ± 0.001	Function impaired
		[mM/sec]	*p* = 0.2067	*p* = 0.2589	*p* = 0.2398	*p* = 0.3427	*p* = 0.4135	*p* = 0.0132	*p* < 0.0001
	**p.Glu384Gly** (c.1151G>C)	HCO_3_^–^/Cl^–^	0.055 ± 0.018 (4)	0.063 ± 0.028 (4)	0.081 ± 0.025 (4)	0.088 ± 0.018 (4)	0.081 ± 0.038 (4)	0.016 ± 0.008	Function impaired
		[nM/sec]	*p* = 0.2875	*p* = 0.407	*p* = 0.7755	*p* = 0.9276	*p* = 0.7679	*p* = 0.066	*p* < 0.0001
		I^–^/Cl^–^	0.008 ± 0.002 (3)	0.009 ± 0.002 (3)	0.016 ± 0.009 (3)	0.013 ± 0.004 (3)	0.012 ± 0.002 (3)	0.002 ± 0.002	Function impaired
		[mM/sec]	*p* = 0.1742	*p* = 0.1931	*p* = 0.3773	*p* = 0.3016	*p* = 0.2592	*p* = 0.1892	*p* < 0.0001
	**p.Ala387Val** (c.1160C>T)	HCO_3_^–^/Cl^–^	0.046 ± 0.027 (4)	0.054 ± 0.14 (4)	0.067 ± 0.032 (4)	0.082 ± 0.053 (4)	0.100 ± 0.051 (4)	0.028 ± 0.011	Function impaired
		[nM/sec]	*p* = 0.2327	*p* = 0.3218	*p* = 0.5252	*p* = 0.8158	*p* = 0.7995	*p* = 0.023	*p* < 0.0001
		I^–^/Cl^–^	0.008 ± 0.005 (3)	0.016 ± 0.005 (3)	0.011 ± 0.002 (3)	0.017 ± 0.008 (3)	0.015 ± 0.003 (3)	0.003 ± 0.002	Function impaired
		[mM/sec]	*p* = 0.1662	*p* = 0.4038	*p* = 0.2439	*p* = 0.4111	*p* = 0.3413	*p* = 0.1819	*p* < 0.0001
	**p.Gly389Arg** (c.1165G>A)	HCO_3_^–^/Cl^–^	0.054 ± 0.049 (3)	0.064 ± 0.023 (3)	0.064 ± 0.015 (3)	0.071 ± 0.013 (3)	0.075 ± 0.023 (3)	0.010 ± 0.009	Function impaired
		[nM/sec]	*p* = 0.3825	*p* = 0.5245	*p* = 0.5265	*p* = 0.6367	*p* = 0.6956	*p* = 0.3057	*p* < 0.0001
		I^–^/Cl^–^	0.011 ± 0.007 (3)	0.009 ± 0.005 (3)	0.012 ± 0.006 (3)	0.011 ± 0.001 (3)	0.013 ± 0.004 (3)	0.001 ± 0.002	Function impaired
		[mM/sec]	*p* = 0.2336	*p* = 0.1889	*p* = 0.2642	*p* = 0.2429	*p* = 0.2878	*p* = 0.4214	*p* < 0.0001
	**p.Gly389Arg** (c.1165G>C)	HCO_3_^–^/Cl^–^	0.057 ± 0.015 (3)	0.110 ± 0.090 (3)	0.076 ± 0.019 (3)	0.077 ± 0.012 (3)	0.113 ± 0.066 (3)	0.015 ± 0.018	Function impaired
		[nM/sec]	*p* = 0.4957	*p* = 0.6993	*p* = 0.7679	*p* = 0.7755	*p* = 0.6584	*p* = 0.408	*p* < 0.0001
		I^–^/Cl^–^	0.018 ± 0.008 (3)	0.014 ± 0.009 (3)	0.013 ± 0.002 (3)	0.015 ± 0.002 (3)	0.013 ± 0.002 (3)	slope < 0	Function impaired
		[mM/sec]	*p* = 0.4607	*p* = 0.3203	*p* = 0.3036	*p* = 0.375	*p* = 0.2997	*p* = 0.4403	*p* < 0.0001
	**p.Ser391Asn** (c.1172G>A)	HCO_3_^–^/Cl^–^	0.076 ± 0.026 (4)	0.084 ± 0.043 (4)	0.075 ± 0.024 (4)	0.085 ± 0.048 (4)	0.066 ± 0.022 (4)	slope < 0	Function impaired
		[nM/sec]	*p* = 0.6844	*p* = 0.8509	*p* = 0.652	*p* = 0.879	*p* = 0.4908	*p* = 0.6983	*p* < 0.0001
		I^–^/Cl^–^	0.012 ± 0.004 (3)	0.015 ± 0.005 (3)	0.018 ± 0.001 (3)	0.019 ± 0.007 (3)	0.024 ± 0.006 (3)	0.005 ± 0.002	Function impaired
		[mM/sec]	*p* = 0.2746	*p* = 0.3545	*p* = 0.462	*p* = 0.4873	*p* = 0.7334	*p* = 0.0053	*p* < 0.0001
	**p.Ser391Arg** (c.1173C>A)	HCO_3_^–^/Cl^–^	0.049 ± 0.009 (3)	0.038 ± 0.015 (3)	0.036 ± 0.039 (3)	0.056 ± 0.022 (3)	0.106 ± 0.074 (3)	0.026 ± 0.014	Function impaired
		[nM/sec]	*p* = 0.3588	*p* = 0.2441	*p* = 0.2289	*p* = 0.4346	*p* = 0.7391	*p* = 0.0897	*p* < 0.0001
		I^–^/Cl^–^	0.014 ± 0.002 (3)	0.014 ± 0.003 (3)	0.015 ± 0.001 (3)	0.013 ± 0.002 (3)	0.013 ± 0.001 (3)	slope < 0	Function impaired
		[mM/sec]	*p* = 0.2964	*p* = 0.3102	*p* = 0.3437	*p* = 0.2906	*p* = 0.2652	*p* = 0.4322	*p* < 0.0001
	**p.Asn392Ser** (c.1175A>G)	HCO_3_^–^/Cl^–^	0.049 ± 0.018 (3)	0.052 ± 0.046 (3)	0.089 ± 0.041 (3)	0.056 ± 0.032 (3)	0.096 ± 0.027 (3)	0.020 ± 0.012	Function impaired
		[nM/sec]	*p* = 0.3348	*p* = 0.3763	*p* = 0.9589	*p* = 0.4347	*p* = 0.9091	*p* = 0.1275	*p* < 0.0001
		I^–^/Cl^–^	0.009 ± 0.001 (3)	0.016 ± 0.003 (3)	0.015 ± 0.002 (3)	0.015 ± 0.001 (3)	0.023 ± 0.005 (3)	0.005 ± 0.001	Function impaired
		[mM/sec]	*p* = 0.1892	*p* = 0.3664	*p* = 0.3464	*p =* 0.340	*p* = 0.6698	*p* = 0.0012	*p* < 0.0001
	**p.Gly396Glu** (c.1187G>A)	HCO_3_^–^/Cl^–^	0.089 ± 0.038 (4)	0.096 ± 0.060 (4)	0.121 ± 0.054 (4)	0.084 ± 0.031 (4)	0.108 ± 0.013 (4)	0.006 ± 0.013	Function impaired
		[nM/sec]	*p* = 0.9512	*p* = 0.9048	*p* = 0.4323	*p* = 0.852	*p* = 0.6605	*p* = 0.6755	*p* < 0.0001
		I^–^/Cl^–^	0.009 ± 0.005 (3)	0.009 ± 0.001 (3)	0.012 ± 0.002 (3)	0.011 ± 0.003 (3)	0.010 ± 0.002 (3)	0.001 ± 0.001	Function impaired
		[mM/sec]	*p* = 0.1751	*p* = 0.1868	*p* = 0.2536	*p =* 0.2304	*p* = 0.2133	*p* = 0.3095	*p* < 0.0001
	**p.Val402Met** (c.1204G>A)	HCO_3_^–^/Cl^–^	0.086 ± 0.026 (3)	0.075 ± 0.010 (3)	0.072 ± 0.029 (3)	0.108 ± 0.071 (3)	0.095 ± 0.020 (3)	0.010 ± 0.013	Function impaired
		[nM/sec]	*p* = 0.9029	*p* = 0.7134	*p* = 0.6663	*p* = 0.6978	*p* = 0.9317	*p* = 0.4368	*p* < 0.0001
		I^–^/Cl^–^	0.013 ± 0.001 (3)	0.011 ± 0.003 (3)	0.016 ± 0.007 (3)	0.018 ± 0.004 (3)	0.018 ± 0.003 (3)	0.004 ± 0.001	Function impaired
		[mM/sec]	*p* = 0.2741	*p* = 0.2302	*p* = 0.3718	*p =* 0.4644	*p* = 0.4446	*p* = 0.0321	*p* < 0.0001
	**p.Thr404Ile** (c.1211C>T)	HCO_3_^–^/Cl^–^	0.091 ± 0.032 (5)	0.062 ± 0.059 (5)	0.085 ± 0.043 (5)	0.094 ± 0.059 (5)	0.100 ± 0.047 (5)	0.010 ± 0.013	Function impaired
		[nM/sec]	*p* = 0.9989	*p* = 0.4374	*p* = 0.870	*p* = 0.940	*p* = 0.8092	*p* = 0.4618	*p* < 0.0001
		I^–^/Cl^–^	0.011 ± 0.004 (3)	0.011 ± 0.004 (3)	0.014 ± 0.004 (3)	0.015 ± 0.003 (3)	0.012 ± 0.007 (3)	0.001 ± 0.002	Function impaired
		[mM/sec]	*p* = 0.2324	*p* = 0.2375	*p* = 0.3348	*p =* 0.365	*p* = 0.2664	*p* = 0.4046	*p* < 0.0001
	**p.Ala406Thr** (c.1216G>A)	HCO_3_^–^/Cl^–^	0.233 ± 0.068 (3)	0.345 ± 0.073 (3)	0.473 ± 0.053 (3)	0.479 ± 0.026 (3)	0.461 ± 0.088 (3)	0.118 ± 0.028	Function impaired
		[nM/sec]	*p* = 0.0172	*p* = 0.0002	*p* < 0.0001	*p* < 0.0001	*p* < 0.0001	*p* = 0.0011	*p* < 0.0001
		I^–^/Cl^–^	0.056 ± 0.009 (3)	0.094 ± 0.021 (3)	0.120 ± 0.037 (3)	0.111 ± 0.040 (3)	0.122 ± 0.033 (3)	0.030 ± 0.011	Function impaired
		[mM/sec]	*p* = 0.3394	*p* = 0.0308	*p* = 0.005	*p =* 0.0095	*p* = 0.0046	*p* = 0.017	*p* = 0.0212
	**p.Ser408Asp** (c.1222_3TC>GA) *	HCO_3_^–^/Cl^–^	0.040 ± 0.019 (4)	0.057 ± 0.020 (4)	0.053 ± 0.012 (4)	0.070 ± 0.031 (4)	0.088 ± 0.031 (4)	0.022 ± 0.007	Function impaired
		[nM/sec]	*p* = 0.1339	*p* = 0.3078	*p* = 0.2554	*p* = 0.5357	*p* = 0.919	*p* = 0.0068	*p* < 0.0001
		I^–^/Cl^–^	0.011 ± 0.002 (3)	0.013 ± 0.003 (3)	0.017 ± 0.004 (3)	0.016 ± 0.007 (3)	0.019 ± 0.002 (3)	0.004 ± 0.001	Function impaired
		[mM/sec]	*p* = 0.2394	*p* = 0.276	*p* = 0.4036	*p =* 0.3921	*p* = 0.4893	*p* = 0.0165	*p* < 0.0001
	**p.Ser408Glu** (c.1222_4TCC>GAG) *	HCO_3_^–^/Cl^–^	0.066 ± 0.032 (5)	0.075 ± 0.021 (5)	0.063 ± 0.034 (5)	0.084 ± 0.059 (5)	0.091 ± 0.057 (5)	0.012 ± 0.012	Function impaired
		[nM/sec]	*p* = 0.477	*p* = 0.6504	*p* = 0.427	*p* = 0.8493	*p* = 0.9929	*p* = 0.3294	*p* < 0.0001
		I^–^/Cl^–^	0.008 ± 0.003 (3)	0.009 ± 0.005 (3)	0.010 ± 0.002 (3)	0.015 ± 0.001 (3)	0.010 ± 0.001 (3)	0.002 ± 0.001	Function impaired
		[mM/sec]	*p* = 0.1561	*p* = 0.1915	*p* = 0.2025	*p =* 0.3308	*p* = 0.2164	*p* = 0.0695	*p* < 0.0001
	**p.Arg409Cys** (c.1225C>T)	HCO_3_^–^/Cl^–^	0.103 ± 0.066 (3)	0.091 ± 0.081 (3)	0.065 ± 0.036 (3)	0.076 ± 0.040 (3)	0.168 ± 0.085 (3)	0.023 ± 0.024	Function impaired
		[nM/sec]	*p* = 0.8259	*p* = 0.9943	*p* = 0.6379	*p* = 0.782	*p* = 0.1685	*p* = 0.3529	*p* < 0.0001
		I^–^/Cl^–^	0.022 ± 0.002 (3)	0.024 ± 0.002 (3)	0.028 ± 0.003 (3)	0.028 ± 0.004 (3)	0.026 ± 0.004 (3)	0.002 ± 0.001	Function impaired
		[mM/sec]	*p* = 0.6456	*p* = 0.7489	*p* = 0.9292	*p =* 0.9227	*p* = 0.8445	*p* = 0.0791	*p* < 0.0001
	**p.Arg409Leu** (c.1226G>T)	HCO_3_^–^/Cl^–^	0.234 ± 0.042 (4)	0.275 ± 0.037 (4)	0.318 ± 0.043 (4)	0.334 ± 0.082 (4)	0.380 ± 0.044 (3)	0.071 ± 0.016	Function impaired
		[nM/sec]	*p* = 0.225	*p* = 0.1221	*p* = 0.0603	*p* = 0.0453	*p* = 0.0003	*p* = 0.0004	*p* < 0.0001
		I^–^/Cl^–^	0.061 ± 0.002 (3)	0.078 ± 0.005 (3)	0.086 ± 0.003 (3)	0.086 ± 0.005 (3)	0.095 ± 0.013 (3)	0.015 ± 0.003	Function impaired
		[mM/sec]	*p* = 0.0552	*p* = 0.0072	*p* = 0.0027	*p =* 0.0024	*p* = 0.009	*p* < 0.0001	*p* = 0.0001
	**p.Ala411Pro** (c.1231G>C)	HCO_3_^–^/Cl^–^	0.064 ± 0.021 (3)	0.084 ± 0.021 (3)	0.071 ± 0.033 (3)	0.066 ± 0.013 (3)	0.065 ± 0.014 (3)	slope < 0	Function impaired
		[nM/sec]	*p* = 0.5137	*p* = 0.8711	*p* = 0.6193	*p* = 0.5427	*p* = 0.529	*p* = 0.6639	*p* < 0.0001
		I^–^/Cl^–^	0.016 ± 0.002 (3)	0.024 ± 0.002 (3)	0.027 ± 0.005 (3)	0.028 ± 0.005 (3)	0.029 ± 0.005 (3)	0.006 ± 0.002	Function impaired
		[mM/sec]	*p* = 0.3929	*p* = 0.7402	*p* = 0.8987	*p =* 0.9163	*p* = 0.9784	*p* = 0.0033	*p* < 0.0001
	**p.Ala411Thr** (c.1231G>A)	HCO_3_^–^/Cl^–^	0.890 ± 0.010 (3)	1.176 ± 0.035 (3)	1.370 ± 0.120 (3)	1.732 ± 0.237 (3)	1.752 ± 0.047 (3)	0.455 ± 0.049	**WT-like**
		[nM/sec]	*p* < 0.0001	*p* < 0.0001	*p* < 0.0001	*p* < 0.0001	*p* < 0.0001	*p* < 0.0001	*p* = 0.0847
		I^–^/Cl^–^	0.064 ± 0.012 (3)	0.079 ± 0.014 (3)	0.094 ± 0.011 (3)	0.113 ± 0.023 (3)	0.123 ± 0.022 (3)	0.030 ± 0.006	Function impaired
		[mM/sec]	*p* = 0.0955	*p* = 0.0224	*p* = 0.0054	*p =* 0.0009	*p* = 0.0004	*p* = 0.0001	*p* = 0.0062
	**p.Val412Ile** (c.1234G>A)	HCO_3_^–^/Cl^–^	0.868 ± 0.186 (3)	1.132 ± 0.503 (3)	1.430 ± 0.706 (3)	1.530 ± 0.532 (3)	2.081 ± 0.849 (3)	0.566 ± 0.194	**WT-like**
		[nM/sec]	*p* = 0.041	*p* = 0.0091	*p* = 0.0016	*p* = 0.0009	*p* < 0.0001	*p* = 0.0121	*p* = 0.8432
		I^–^/Cl^–^	0.085 ± 0.009 (3)	0.117 ± 0.005 (3)	0.122 ± 0.016 (3)	0.141 ± 0.015 (3)	0.146 ± 0.006 (3)	0.029 ± 0.004	Function impaired
		[mM/sec]	*p* = 0.005	*p* = 0.0002	*p* < 0.0001	*p* < 0.0001	*p* < 0.0001	*p* < 0.0001	*p* = 0.0036
	**p.Gln413Pro** (c.1238A>C)	HCO_3_^–^/Cl^–^	0.095 ± 0.021 (3)	0.081 ± 0.027 (3)	0.090 ± 0.057 (3)	0.144 ± 0.056 (3)	0.121 ± 0.100 (3)	0.023 ± 0.020	Function impaired
		[nM/sec]	*p* = 0.9349	*p* = 0.8519	*p* = 0.984	*p* = 0.3123	*p* = 0.5596	*p* = 0.2745	*p* < 0.0001
		I^–^/Cl^–^	0.019 ± 0.003 (3)	0.026 ± 0.007 (3)	0.027 ± 0.002 (3)	0.024 ± 0.001 (3)	0.022 ± 0.004 (3)	0.001 ± 0.002	Function impaired
		[mM/sec]	*p* = 0.4857	*p* = 0.8454	*p* = 0.8783	*p* = 0.7352	*p* = 0.6318	*p* = 0.6167	*p* < 0.0001
	**p.Gln413Arg** (c.1238A>G)	HCO_3_^–^/Cl^–^	0.313 ± 0.107 (3)	0.349 ± 0.156 (3)	0.440 ± 0.212 (3)	0.459 ± 0.203 (3)	0.424 ± 0.208 (3)	0.066 ± 0.059	Function impaired
		[nM/sec]	*p* = 0.0652	*p* = 0.0349	*p* = 0.0069	*p* = 0.0048	*p* = 0.0092	*p* = 0.2848	*p* < 0.0001
		I^–^/Cl^–^	0.039 ± 0.011 (3)	0.050 ± 0.009 (3)	0.058 ± 0.002 (3)	0.064 ± 0.002 (3)	0.072 ± 0.007 (3)	0.016 ± 0.002	Function impaired
		[mM/sec]	*p* = 0.5267	*p* = 0.1857	*p* = 0.0756	*p* = 0.0391	*p* = 0.0139	*p* < 0.0001	*p* = 0.0001
	**p.Glu414Lys** (c.1240G>A)	HCO_3_^–^/Cl^–^	0.305 ± 0.075 (3)	0.314 ± 0.166 (3)	0.327 ± 0.079 (3)	0.404 ± 0.106 (3)	0.429 ± 0.262 (3)	0.067 ± 0.050	Function impaired
		[nM/sec]	*p* = 0.0431	*p* = 0.0356	*p* = 0.0279	*p* = 0.0055	*p* = 0.0033	*p* = 0.1998	*p* < 0.0001
		I^–^/Cl^–^	0.040 ± 0.008 (3)	0.045 ± 0.007 (3)	0.049 ± 0.002 (3)	0.051 ± 0.005 (3)	0.061 ± 0.008 (3)	0.010 ± 0.002	Function impaired
		[mM/sec]	*p* = 0.4681	*p* = 0.3109	*p* = 0.2121	*p* = 0.1617	*p* = 0.0545	*p* = 0.0007	*p* < 0.0001
	**p.Ser415Gly** (c.1243A>G)	HCO_3_^–^/Cl^–^	0.384 ± 0.171 (3)	0.484 ± 0.075 (3)	0.727 ± 0.184 (3)	0.824 ± 0.260 (3)	0.774 ± 0.260 (3)	0.224 ± 0.070	Function impaired
		[nM/sec]	*p* = 0.0303	*p* = 0.0059	*p* = 0.0001	*p* < 0.0001	*p* < 0.0001	*p* = 0.0072	*p* < 0.0001
		I^–^/Cl^–^	0.068 ± 0.011 (3)	0.089 ± 0.011 (3)	0.097 ± 0.014 (3)	0.107 ± 0.016 (3)	0.112 ± 0.007 (3)	0.021 ± 0.004	Function impaired
		[mM/sec]	*p* = 0.0377	*p* = 0.0039	*p* = 0.0017	*p* = 0.0006	*p* = 0.0003	*p* = 0.0002	*p* = 0.0007
	**p.Ser415Arg** (c.1245C>A)	HCO_3_^–^/Cl^–^	0.634 ± 0.135 (3)	0.923 ± 0.072 (3)	0.915 ± 0.013 (3)	1.084 ± 0.041 (3)	1.242 ± 0.057 (3)	0.275 ± 0.033	Function impaired
		[nM/sec]	*p* < 0.0001	*p* < 0.0001	*p* < 0.0001	*p* < 0.0001	*p* < 0.0001	*p* < 0.0001	*p* < 0.0001
		I^–^/Cl^–^	0.052 ± 0.009 (3)	0.069 ± 0.018 (3)	0.080 ± 0.011 (3)	0.094 ± 0.010 (3)	0.091 ± 0.013 (3)	0.021 ± 0.004	Function impaired
		[mM/sec]	*p* = 0.203	*p* = 0.0371	*p* = 0.0102	*p* = 0.0023	*p* = 0.0033	*p* = 0.0005	*p* = 0.0006
	**p.Ser532Ile** (c.1595G>T)	HCO_3_^–^/Cl^–^	0.549 ± 0.091 (4)	0.793 ± 0.254 (4)	0.816 ± 0.107 (4)	0.845 ± 0.267 (4)	0.912 ± 0.245 (4)	0.155 ± 0.063	Function impaired
		[nM/sec]	*p* = 0.0011	*p* < 0.0001	*p* < 0.0001	*p* < 0.0001	*p* < 0.0001	*p* = 0.0244	*p* < 0.0001
		I^–^/Cl^–^	0.053 ± 0.011 (3)	0.055 ± 0.011 (3)	0.069 ± 0.018 (3)	0.086 ± 0.028 (3)	0.095 ± 0.009 (3)	0.023 ± 0.006	Function impaired
		[mM/sec]	*p* = 0.2373	*p* = 0.198	*p* = 0.0572	*p* = 0.0108	*p* = 0.0045	*p* = 0.0011	*p* = 0.0015
** Figure 5 **	**p.Ala104Val** (c.311C>T)	HCO_3_^–^/Cl^–^	0.079 ± 0.017 (4)	0.152 ± 0.080 (4)	0.110 ± 0.035 (4)	0.132 ± 0.029 (4)	0.105 ± 0.039 (4)	0.006 ± 0.015	Function impaired
		[nM/sec]	*p* = 0.7598	*p* = 0.1325	*p* = 0.6295	*p* = 0.3075	*p* = 0.718	*p* = 0.7015	*p* < 0.0001
	**p.Ala104Thr** (c.310G>A) *	HCO_3_^–^/Cl^–^	0.363 ± 0.051 (6)	0.402 ± 0.075 (6)	0.468 ± 0.039 (6)	0.528 ± 0.100 (6)	0.443 ± 0.084 (6)	0.057 ± 0.022	Function impaired
		[nM/sec]	*p* < 0.0001	*p* < 0.0001	*p* < 0.0001	*p* < 0.0001	*p* < 0.0001	*p* = 0.0159	*p* < 0.0001
	**p.Ala451Gly** (c.1352C>G) *	HCO_3_^–^/Cl^–^	0.479 ± 0.092 (3)	0.911 ± 0.063 (3)	0.993 ± 0.129 (3)	1.284 ± 0.089 (3)	1.836 ± 0.184 (3)	0.618 ± 0.056	**WT-like**
		[nM/sec]	*p* = 0.0062	*p* < 0.0001	*p* < 0.0001	*p* < 0.0001	*p* < 0.0001	*p* < 0.0001	*p* = 0.7423
	**p.Ala451Ser** (c.1351G>T) *	HCO_3_^–^/Cl^–^	0.572 ± 0.140 (3)	0.921 ± 0.162 (3)	1.168 ± 0.127 (3)	1.406 ± 0.167 (3)	1.749 ± 0.251 (3)	0.568 ± 0.057	**WT-like**
		[nM/sec]	*p* = 0.0027	*p* < 0.0001	*p* < 0.0001	*p* < 0.0001	*p* < 0.0001	*p* < 0.0001	*p* = 0.7667
	**p.Ala451Leu** (c.1351_2GC>CT) *	HCO_3_^–^/Cl^–^	0.097 ± 0.047 (3)	0.144 ± 0.013 (3)	0.100 ± 0.036 (3)	0.093 ± 0.014 (3)	0.143 ± 0.064 (3)	0.008 ± 0.015	Function impaired
		[nM/sec]	*p* = 0.3455	*p* = 0.0283	*p* = 0.3058	*p* = 0.4034	*p* = 0.0299	*p* = 0.6021	*p* < 0.0001

Bold fonts were used to emphasize the amino acid changes of the variants from nucleotide changes in the parentheses.

**Figure 3 ijms-25-02759-f003:**
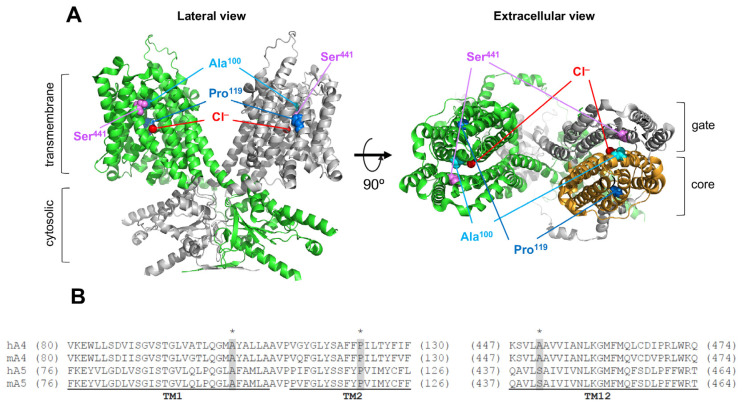
Ala^100^, Pro^119^, and Ser^441^ sites in prestin. (**A**) The homodimeric structure of human prestin (PDB: 7LGU). Protomers are shown in green and gray. In right panel, the core domain of one of the protomers is shown in bright orange. The Ala^100^, Pro^119^, and Ser^441^ sites and bound chlorides are indicated by cyan, blue, purple, and red spheres, respectively. (**B**) Partial amino acid sequences of human and mouse pendrin (A4) and prestin (A5). Numbers in parentheses indicate the residue numbers at the N- and C-terminal ends. The residues with asterisks indicate Ala^100^, Pro^119^, and Ser^441^ in human prestin (hA5) and equivalents in others.

**Figure 4 ijms-25-02759-f004:**
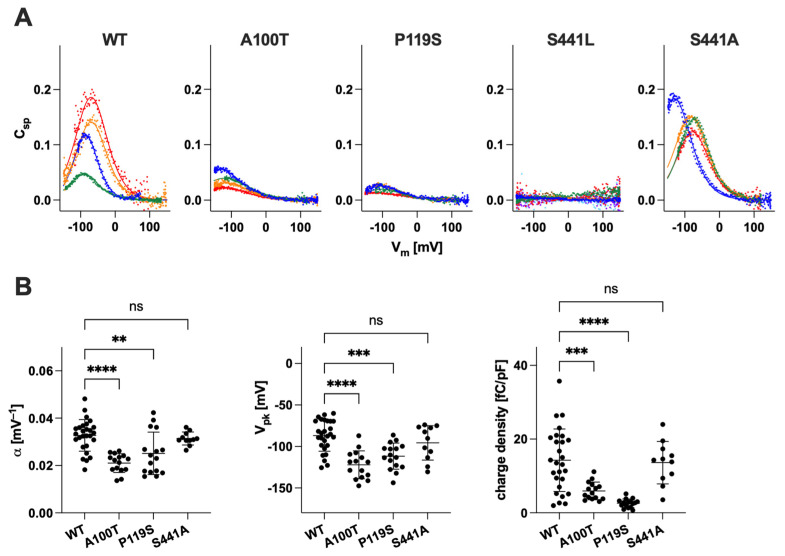
NLC measurements. (**A**) Examples of NLC recorded in HEK293T cells expressing WT, p.A100T, p.P119S, p.S441L, or p.S441A prestin. Different colors indicate individual recordings. (**B**) Summaries of the NLC parameters (α, V_pk_, and CD). Error bars indicate SD. ns, *p* ≥ 0.05; ** 0.001 < *p* ≤ 0.01; *** 0.0001 < *p* ≤ 0.001; **** *p* ≤ 0.0001.

**Figure 5 ijms-25-02759-f005:**
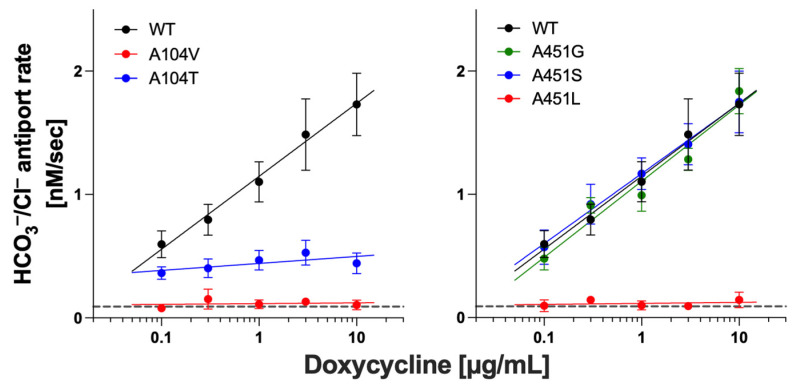
The effects of missense changes at Ala^104^ and Ala^451^ on the anion transport function of pendrin. HCO_3_^−^/Cl^−^ antiport assay conducted for p.A104V (**left**), p.A104T (**left**), p.A451G (**right**), p.A451S (**right**), and p.A451L (**right**) pendrin alongside WT control. Error bars indicate SD. Horizontal dotted lines indicate transport rates of uninduced cells. Solid lines indicate linear regressions (log_10_ [Dox] vs. transport rates). Sample size information and statistics are provided in Table 1.

### 2.3. Comparison to AlphaMissense Variant Effect Predictor

The pathogenicity of disease-associated variants is best assessed by functional analyses such as those presented here; however, experimental characterization of variants is laborious and remains reactive to the rapid identification of novel variants. Recent advancements in computational methods using machine learning have prompted us to evaluate the performance of AlphaMissense (AM), a novel tool for predicting the pathogenicity of missense variants [16]. First, we visualized the provided AM prediction scores of pendrin and prestin at every amino acid position as heatmaps (Appendix A). AM pathogenicity predictions for both pendrin and prestin show similarity, such as (i) variants in regions that lack structural information (C- and N-termini, IDR) tend to be benign (i.e., predominantly “blue”); and (ii) variants in transmembrane regions tend to be less tolerant to missense changes (i.e., predominantly “red”) in the heatmap (Appendix A). These observations are in line with the fact that AM utilizes structural information, and both pendrin and prestin share quite similar overall molecular architecture. To evaluate the accuracy of AM, we plotted the HCO_3_^−^/Cl^−^ antiporter activities of pendrin variants measured in our previous study [10] and this study, which are normalized to WT control, against the AM pathogenicity scores (Figure 6). We found an overall correlation between our functional assay data and AM prediction scores of r = −0.6497 (nonparametric Spearman correlation), which is in line with the example given in their report (GCK, r = −0.65, Figure 3G in [16]). Many variants tested in this study are clustered around highly conserved TM9-10 that are essential for the transport function, and thus many had little or no activities, skewing the distribution of variants (Figure 6, inset). In addition, we found several variants that were misclassified by AM as benign when there is vastly reduced activity (for example, p.N392S) or vice versa (ambiguous/pathogenic when their activity is WT-like, such as p.T307A, p.L117F, and p.F354S) (Table 2). For prestin variants, AM scores for p.A100T, p.P119S, and p.S441L were 0.9138, 0.7712, and 0.916, respectively, all predicted to be pathogenic.

## 3. Discussion

Precisely understanding the relationships between genotypes and disease phenotypes is the holy grail of the field of human genetics. Although sequencing information becomes more and more accessible, assigning pathogenicity to variations of sequences remains challenging due to the complexity of determining the causality of variants to disease states [69]. Although experimental interrogation is ideal, our collective experimental efforts have only scratched the surface of building an atlas of variant effects in the genome. Consequently, the vast majority of disease-associated variants, including those of pendrin and prestin, remain as variants of unknown significance (VUS) [70]. To bridge the gap, in silico tools have been developed to predict whether a variant plays a causative role in a disease state. Large numbers of variant effect predictors (VEPs) have been reported over the years using various approaches with various degrees of performance [71]. For the deafness genes, a measure for protein-folding stability was used to re-classify VUSs in patients due to protein misfolding [72]. Recently developed AlphaMissense (AM) also uses structural information, although it does not predict protein stability. Rather, it classifies missense variants as benign, ambiguous, or pathogenic based on their prediction scores [16]. As the study provided AM prediction across the entire human proteome, we compared the available AM prediction scores for pendrin to our experimental results on quantitated transport activities. We found a good correlation (r = −0.6497, Figure 6), demonstrating their utility in informing the variant effects. It must be noted, however, that the uncertainty still exists as several pendrin variants were misclassified. Thus, the usage of VEPs, even the one that demonstrated superior performance over many existing VEPs across multiple benchmarks as AM, must exercise the utmost caution in clinical settings.

Recent efforts in large-scale systemic studies (multiplexed assay of variant effects, MAVE) have reported functional consequences of hundreds of thousands of variants across genome regions to date [73,74]. For proteins such as pendrin and prestin, however, large-scale multiplexing assays will be challenging as their dysfunctions may not result in easily discernible cellular phenotypes. Nevertheless, our small-scale in vitro assays are robust in determining the degrees of functional impairment of the variants found in human patients. The large variation in PDS and DFNB4 phenotypes suggests large variations in the activities of pendrin variants, thus warranting an experimental approach that can detect a subtle difference in transport activities. One caveat will be that our approach to heterologous expression systems may not be able to address the effects of variants on protein expression levels in the host environment, as we forcibly induce protein expression by doxycycline from cDNAs for our transporter assays. Also not addressed here are the variant effects on protein–protein interactions. For example, the intrinsically disordered region (IDR) within the cytosolic domain of prestin mediates the binding of calmodulin, thereby allowing modulation of protein activity by calcium, which is likely conserved in other SLC26 proteins, including pendrin [75]. Additionally, IQ-motif containing GTPase-activating protein 1 (IQGAP1) was shown to interact with the C-terminal region of pendrin and enhance its transport activity in kidney cells [76]. Interestingly, co-expression of IQGAP1 and pendrin also enhanced the plasma membrane (PM) localization of pendrin, suggesting another layer of regulatory interaction. It is curious that although both pendrin and prestin are conserved and the handful of disease-associated variants in prestin also similarly affected pendrin in this study, prestin is targeted to the lateral membrane in the outer hair cells (OHCs), while pendrin is targeted to the apical membranes of epithelial cells in the thyroid and in the inner ear [77,78,79,80]. Thus, it will require additional considerations such as co-expression of interacting partners or using the host cell lines in order to interrogate the effects on variants found in regions where such regulatory interactions were reported in the future.

Although functional annotation of deafness-associated variations in the genome may not be complete until the hearing phenotype is assessed in vivo, our continuing efforts to characterize missense variants in vitro provide invaluable information regarding the variant effects, which can be used to prioritize future efforts at the organismal level. With ever-improving VEPs such as AM, future studies will provide clinically useful information of variant effects with more confidence.

## 4. Materials and Methods

### 4.1. Generation of Stable Cell Lines

cDNAs coding human pendrin (NM_000441.2) and gerbil prestin (AF230376.2), (WT and variants) with a mTurquoise (mTq2) tag on the C-terminus were cloned into a pSBtet-pur vector (Addgene, Watertown, NY, USA) using SfiI sites. Stable cell lines carrying these constructs were established in HEK293T cells as previously described [10]. Stable cells were maintained in DMEM supplemented with 10% FBS and 1 µg/mL puromycin (Themo Fisher Scientific, Waltham, MA, USA), and the expression of the pendrin and prestin constructs were induced by the addition of doxycycline (Dox) in the media. For I^−^/Cl^−^ antiport assay, the pendrin-expressing pSBtet-Pur vectors were transfected in the HEK293T cell line that constitutively express mVenus^p.H148Q/p.I152L^ as previously described [10].

### 4.2. Fluorometric Anion Transport Assays

Fluorometric HCO_3_^−^/Cl^−^ and I^−^/Cl^−^ antiport assays were established and described in detail in a previous study [10]. Briefly, for HCO_3_^−^/Cl^−^ antiport assay, stable HEK293T cells expressing pendrin constructs were loaded with a pH indicator, SNARF-5F (S23923, Thermo Fisher Scientific, Waltham, MA, USA) in a high chloride buffer containing (mM): 140 NaCl, 4.5 KCl, 1 MgCl_2_, 2.5 CaCl_2_, 20 HEPES (pH 7.4). The antiport assay was initiated by an automated injection of a low chloride buffer containing (mM): 125 Na-gluconate, 5 K-gluconate, 1 MgCl_2_, 1 CaCl_2_, 20 HEPES, 25 NaHCO_3_ with 5% CO_2_ in Synergy Neo2 (Agilent/BioTek, Santa Clara, CA, USA). The fluorescence of SNARF-5F were measured in a time dependent manner using Synergy Neo2 (Agilent/BioTek) and the data analyzed offline as described previously [10]. For I^−^/Cl^−^ antiport assay, cells expressing both pendrin variants and iodide sensitive fluorescent protein, mVenus^p.H148Q/p.I152L^, were resuspended in 200 μL of a high Cl^−^ buffer containing (mM): 150 NaCl, 1 MgCl_2_, 1 CaCl_2_, 20 HEPES, (pH 7.5). The I^−^/Cl^−^ antiport assay using 160 µL of the cell suspension was initiated by an automated injection of 80 μL of a high I^−^ buffer containing (mM): 150 NaI, 1 MgCl_2_, 1 CaCl_2_, 20 HEPES (pH 7.5) in Synergy Neo2 (Agilent/BioTek). The fluorescence intensities of mVenus^p.H148Q/p.I152L^ and mTq2 were simultaneously measured in a time-dependent manner using Synergy Neo2 (Agilent/BioTek) and the data analyzed offline as described previously [10].

### 4.3. Whole-Cell Recordings

Whole-cell recordings were performed at room temperature using the Axopatch 200B amplifier (Molecular Devices, San Jose, CA, USA) with a 10 kHz low-pass filter. Recording pipettes pulled from borosilicate glass were filled with an ionic blocking intracellular solution containing (mM): 140 CsCl, 2 MgCl_2_, 10 EGTA, and 10 HEPES (pH 7.4). Cells were bathed in an extracellular solution containing (mM): 120 NaCl, 20 TEA-Cl, 2 CoCl_2_, 2 MgCl_2_, 10 HEPES (pH 7.4). Osmolality was adjusted to 309 mOsmol/kg with glucose. Holding potentials were set to 0 mV. NLC was measured using sinusoidal voltage stimuli (2.5-Hz, 120 or 150 mV amplitude) superimposed with two higher frequency stimuli (390.6 and 781.2 Hz, 10 mV amplitude). Data were collected by jClamp (SciSoft Company, New Haven, CT, USA) [81].

### 4.4. NLC Data Analysis

Voltage-dependent *C_m_* data were analyzed using the following two-state Boltzmann equation:(1)$$C_{m}=\frac{\alpha Q_{max}exp[\alpha \left(V_{m}-V_{pk}\right)]}{{\left\{1+exp\left[\alpha \left(V_{m}-V_{pk}\right)\right]\right\}}^{2}}+C_{lin}$$
where *α* is the slope factor of the voltage-dependence of charge transfer, *Q_max_* is the maximum charge transfer, *V_m_* is the membrane potential, *V_pk_* is the voltage at which the maximum charge movement is attained, and *C_lin_* is the linear capacitance. The specific capacitance, *C_sp_*, was calculated as (*C_m_* − *C_lin_*)/*C_lin_*.

### 4.5. Statistical Analyses

Statistical analyses for fluorometric antiport assays were performed as previously described [10]. Briefly, five different Dox dosage conditions were compared to the rates of uninduced cells by one-way ANOVA followed by an uncorrected Fisher’s Least Significant Difference (LSD). To assess the dependency of the transport rates on Dox dosage, linear regressions (log_10_ [Dox] vs. transport rates) were performed. *F* tests were performed to find the difference in the Dox-dependence between WT versus pendrin variants. To obtain normalized activities for each pendrin variant, the slope value from the linear regressions described above was divided by that of WT. The uncertainties (*σ*) associated with division computations were calculated as previously described [82] using the following equation:(2)$$\sigma_{A/B}=\left|\frac{A}{B}\right|\sqrt{{\left(\frac{\sigma_{A}}{A}\right)}^{2}+{\left(\frac{\sigma_{B}}{B}\right)}^{2}}$$
where *A* and *B* are the mean values with associated errors in linear regressions, *σ_A_* and *σ_B_*, respectively. These values are reported as errors in Table 2 under “%WT Activity”. The Spearman Correlation coefficient (r) between pendrin HCO_3_^−^/Cl^−^ transport activities and the AM scores was calculated using Prism 10.2.0. For NLC data analyses, a one-way ANOVA combined with Tukey’s post hoc test was used for multiple comparisons. In all statistical tests, *p* < 0.05 was considered statistically significant.

## Figures and Tables

**Figure 1 ijms-25-02759-f001:**
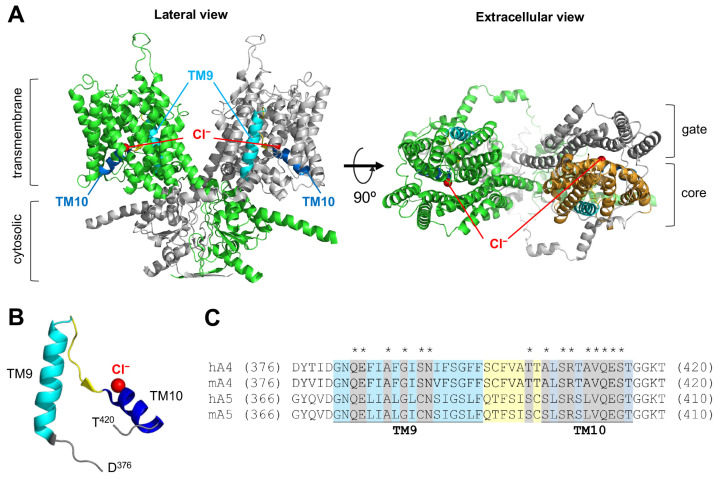
TM9-10 of pendrin. (**A**) The homodimeric structure of mouse pendrin (PDB: 7WK1). Protomers are shown in green and gray. Transmembrane and cytosolic domains are indicated in the lateral view (left). TM9 and TM10 are highlighted in cyan and blue, respectively, with connecting residues highlighted in yellow. The bound chlorides are indicated by red spheres. In the extracellular view (right), the core domain of one of the protomers is shown in bright orange. (**B**) TM9 and TM10 region of the structure (residues 376–420), extracted from the right protomer in (**A**). TM9 (381–398) is shown in cyan, linker region (399–405) is in yellow, and TM10 (406–416) is in blue. Bound chloride is shown as a red sphere. (**C**) Partial amino acid sequences of human and mouse pendrin (A4) and prestin (A5) showing the TM9 and TM10 region. Numbers in parentheses indicate the residue numbers at the N- and C-terminal ends of the partial amino acid sequences. TM9 and TM10 are highlighted in cyan and blue, respectively, with connecting residues highlighted in yellow as in (**A**). The residues with asterisks (*) on top and gray shades indicate the locations of missense changes evaluated in Figure 2.

**Figure 2 ijms-25-02759-f002:**
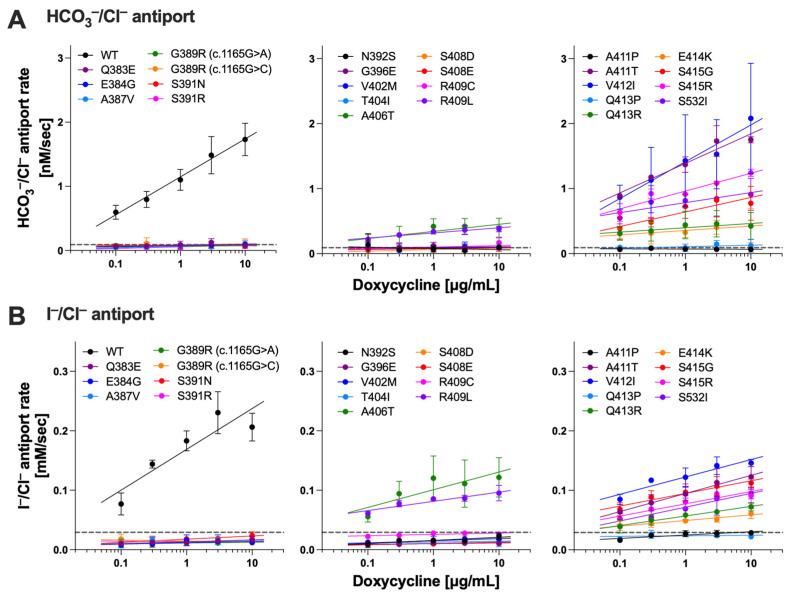
HCO_3_^−^/Cl^−^ and I^−^/Cl^−^ antiport assays on pendrin variants. HCO_3_^−^/Cl^−^ (**A**) and I^−^/Cl^−^ (**B**) antiport rates were plotted against doxycycline (Dox) concentration (0.1–10 µg/mL) for each mTq2-tagged pendrin variant alongside WT as indicated. Horizontal dotted lines indicate transport rates of uninduced cells. Error bars indicate SD. Solid lines indicate linear regressions (log_10_ [Dox] vs. transport rates). Sample size information and statistics are provided in Table 1.

**Figure 6 ijms-25-02759-f006:**
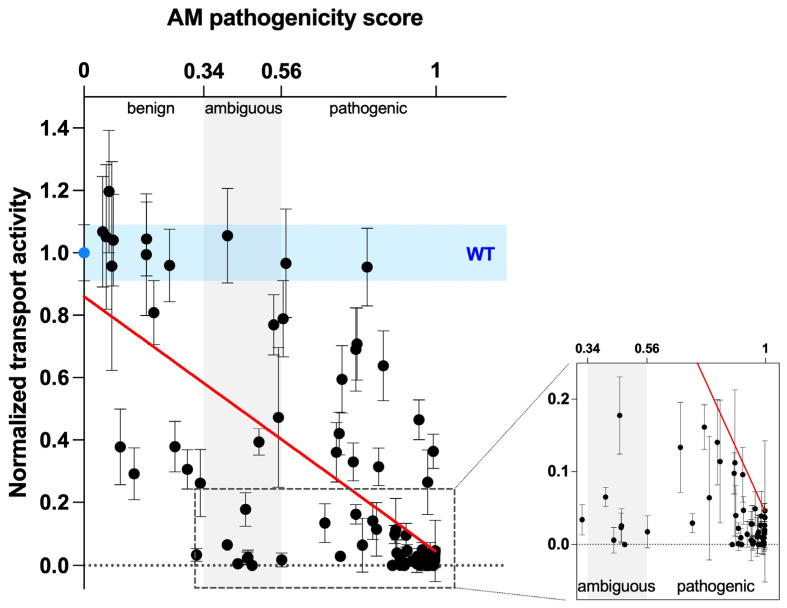
Correlation with AlphaMissense pathogenicity scores. HCO_3_^−^/Cl^−^ transport rates of pendrin WT and variants from this study and the previous study [10] were normalized to the WT value and plotted against AlphaMissense (AM) pathogenicity scores [16]. Light blue shades indicate HCO_3_^−^/Cl^−^ rate of WT with errors (propagated errors). Gray shade between AM scores 0.34–0.56 marks the “ambiguous” category, with scores lower being “benign” and higher being “pathogenic” as indicated. Red line indicates the linear fit between the HCO_3_^−^/Cl^−^ antiport activity and the AM scores. Inset: Region indicated by the broken lines in left are enlarged to visualize variants with little or no HCO_3_^−^/Cl^−^ transport activity.

**Table 2 ijms-25-02759-t002:** Comparison of experimental data vs. AM scores. The slope values of the Dox-dependent HCO_3_^−^/Cl^−^ antiport activities from our previous study [10] and from this study for each variant were normalized to WT. For variants with negative slope values (slope < 0), normalized activity of 0 was assigned (highlighted in yellow) with no error propagations. Variants with WT-like HCO_3_^−^/Cl^−^ antiport activity (*p* ≥ 0.05) were shaded in light blue, and ones with impaired functions (*p* < 0.05) are shaded in pale pink. AM scores for pathogenicity classification uses the same color shades as in Appendix A. Asterisks (*) indicate variants not found in human patients (as of January 2024).

			HCO_3_^–^/Cl^–^ Antiport Assay Results		AM Prediction	
			%WT Activity	Variant Effect	AM Score	AM Category
**Wasano et al., 2020 [10]**	**p.Ser28Gly** (c.82A>G)		29.2 ± 8.3	Function impaired	0.1421	benign
	**p.Ser49Arg** (c.147C>G)		104.1 ± 13.7	WT-like	0.0831	benign
	**p.Pro76Ser** (c.226C>T)		30.5 ± 6.3	Function impaired	0.2937	benign
	**p.Ser90Leu** (c.269C>T)		0.1 ± 0.9	Function impaired	0.8985	pathogenic
	**p.Thr99Arg** (c.296C>G)		0	Function impaired	0.9802	pathogenic
	**p.Leu117Phe** (c.349C>T)		95.5 ± 12.5	WT-like	0.804	pathogenic
	**p.Pro123Ser** (c.367C>T)		2.9 ± 1.3	Function impaired	0.7286	pathogenic
	**p.Gly131Val** (c.392G>T)		0.4 ± 1.7	Function impaired	0.9925	pathogenic
	**p.Ser133Thr** (c.397T>A)		2.3 ± 2.0	Function impaired	0.4642	ambiguous
	**p.Gly139Ala** (c.416G>C)		0	Function impaired	0.9105	pathogenic
	**p.Met147Thr** (c.440T>C)		0.6 ± 2.8	Function impaired	0.9459	pathogenic
	**p.Met147Val** (c.439A>G)		6.4 ± 8.5	Function impaired	0.7909	pathogenic
	**p.Val163Ile** (c.487G>A)		105.1 ± 23.2	WT-like	0.0632	benign
	**p.Val186Phe** (c.556G>T)		0	Function impaired	0.4777	ambiguous
	**p.Thr193Ile** (c.578C>T)		0.9 ± 1.3	Function impaired	0.9082	pathogenic
	**p.Tyr214Cys** (c.641A>G)		31.4 ± 6.0	Function impaired	0.8371	pathogenic
	**p.Val239Asp** (c.716T>A)		4.9 ± 2.3	Function impaired	0.9612	pathogenic
	**p.Asp266Asn** (c.796G>A)		106.7 ± 17.7	WT-like	0.0528	ambiguous
	**p.Thr307Ala** (c.919A>G)		105.5 ± 15.2	WT-like	0.4074	ambiguous
	**p.Asn324Tyr** (c.970A>T)		99.4 ± 19.5	WT-like	0.1773	pathogenic
	**p.Gly334Val** (c.1001G>T)		59.5 ± 10.8	Function impaired	0.7329	pathogenic
	**p.Phe354Ser** (c.1061T>C)		96.6 ± 17.4	WT-like	0.5739	pathogenic
	**p.Lys369Glu** (c.1105A>G)		39.4 ± 4.2	Function impaired	0.4968	ambiguous
	**p.Ala372Val** (c.1115C>T)		0.1± 0.9	Function impaired	0.9528	pathogenic
	**p.Asn392Tyr** (c.1174A>T)		0.5 ± 0.7	Function impaired	0.9493	pathogenic
	**p.Ser399Pro** (c.1195T>C)		78.9 ± 12.2	Function impaired	0.5658	pathogenic
	**p.Ser408Phe** (c.1223C>T)		0.9 ± 1.3	Function impaired	0.9938	pathogenic
	**p.Arg409His** (c.1226G>A)		2.8 ± 2.6	Function impaired	0.9499	pathogenic
	**p.Thr410Lys** (c.1229C>A)		1.9 ± 1.1	Function impaired	0.9957	pathogenic
	**p.Thr410Met** (c.1229C>T)		1.4 ± 1.7	Function impaired	0.9314	pathogenic
	**p.Thr416Pro** (c.1246A>C)		2.2 ± 0.7	Function impaired	0.8985	pathogenic
	**p.Gln421Leu** (c.1262A>T)		14.1 ± 5.9	Function impaired	0.8205	pathogenic
	**p.Ile426Asn** (c.1277T>A)		63.8 ± 11.2	Function impaired	0.8501	pathogenic
	**p.Leu445Trp** (c.1334T>G)		0	Function impaired	0.9933	pathogenic
	**p.Asn457Lys** (c.1371C>A)		36.4 ± 5.5	Function impaired	0.992	pathogenic
	**p.Arg470His** (c.1409G>A)		119.6 ± 19.7	WT-like	0.0709	benign
	**p.Val483Glu** (c.1448T>A)		70.7 ± 11.6	WT-like	0.7757	pathogenic
	**p.Gly497Ser** (c.1489G>A)		9.8 ± 2.9	Function impaired	0.883	pathogenic
	**p.Thr527Pro** (c.1579A>C)		42.0 ± 6.8	Function impaired	0.7253	pathogenic
	**p.Ile529Ser** (c.1586T>G)		16.1 ± 3.1	Function impaired	0.7726	pathogenic
	**p.Tyr556Cys** (c.1667A>G)		6.5 ± 1.3	Function impaired	0.4069	ambiguous
	**p.Cys565Tyr** (c.1694G>A)		80.8 ± 10.3	WT-like	0.1986	benign
	**p.Ser657Asn** (c.1970G>A)		33.0 ± 6.0	Function impaired	0.7648	pathogenic
	**p.Val659Leu** (c.1975G>C)		37.9 ± 8.1	Function impaired	0.2582	benign
	**p.Ser666Phe** (c.1997C>T)		17.8 ± 5.3	Function impaired	0.4588	ambiguous
	**p.Asp669Glu** (c.2007C>A)		0.0 ± 0.0	Function impaired	0.973	pathogenic
	**p.Phe683Ser** (c.2048T>C)		2.8 ± 1.8	Function impaired	0.9461	pathogenic
	**p.Phe692Leu** (c.2074T>C)		36.0 ± 9.6	Function impaired	0.7169	pathogenic
	**p.Leu703Pro** (c.2108T>C)		0.3 ± 0.9	Function impaired	0.955	pathogenic
	**p.Thr721Met** (c.2162C>T)		0.6 ± 1.7	Function impaired	0.4373	ambiguous
	**p.His723Arg** (c.2168A>G)		13.4 ± 6.2	Function impaired	0.684	pathogenic
**This study**	**p.Gln383Glu** (c.1147C>G)		2.5 ± 2.4	Function impaired	0.4658	ambiguous
	**p.Glu384Gly** (c.1151G>C)		2.7 ± 1.4	Function impaired	0.9808	pathogenic
	**p.Ala387Val** (c.1160C>T)		4.7 ± 1.9	Function impaired	0.9179	pathogenic
	**p.Gly389Arg** (c.1165G>A)		1.7 ± 1.5	Function impaired	0.996	pathogenic
	**p.Gly389Arg** (c.1165G>C)		2.5 ± 3.1	Function impaired	0.996	pathogenic
	**p.Ser391Asn** (c.1172G>A)		0	Function impaired	0.8762	pathogenic
	**p.Ser391Arg** (c.1173C>A)		4.3 ± 2.4	Function impaired	0.9974	pathogenic
	**p.Asn392Ser** (c.1175A>G)		3.4 ± 2.0	Function impaired	0.3199	benign
	**p.Gly396Glu** (c.1187G>A)		1.0 ± 0.2	Function impaired	0.9866	pathogenic
	**p.Val402Met** (c.1204G>A)		1.7 ± 2.2	Function impaired	0.5617	pathogenic
	**p.Thr404Ile** (c.1211C>T)		1.7 ± 2.2	Function impaired	0.9734	pathogenic
	**p.Ala406Thr** (c.1216G>A)		20.0 ± 4.9	Function impaired	0.8845	pathogenic
	**p.Ser408Asp** (c. 1222_3TC>GA) *		3.7 ± 1.2	Function impaired	0.9971	pathogenic
	**p.Ser408Glu** (c.1222_4TCC>GAG) *		2.0 ± 2.0	Function impaired	0.9964	pathogenic
	**p.Arg409Cys** (c.1225C>T)		3.4 ± 4.1	Function impaired	0.8893	pathogenic
	**p.Arg409Leu** (c.1226G>T)		12.0 ± 2.8	Function impaired	0.9763	pathogenic
	**p.Ala411Pro** (c.1231G>C)		0	Function impaired	0.9893	pathogenic
	**p.Ala411Thr** (c.1231G>A)		77.0 ± 9.8	WT-like	0.5393	ambiguous
	**p.Val412Ile** (c.1234G>A)		95.8 ± 33.5	WT-like	0.0792	benign
	**p.Gln413Pro** (c.1238A>C)		3.9 ± 3.4	Function impaired	0.9839	pathogenic
	**p.Gln413Arg** (c.1238A>G)		11.2 ± 10.0	Function impaired	0.8859	pathogenic
	**p.Glu414Lys** (c.1240G>A)		11.3 ± 8.5	Function impaired	0.8309	pathogenic
	**p.Ser415Gly** (c.1243A>G)		37.9 ± 12.1	Function impaired	0.1027	benign
	**p.Ser415Arg** (c.1245C>A)		46.5 ± 6.4	Function impaired	0.9514	pathogenic
	**p.Ser532Ile** (c.1595G>T)		26.2 ± 10.8	Function impaired	0.3304	benign
	**p.Ala104Val** (c.311C>T)		1.0 ± 2.5	Function impaired	0.9684	pathogenic
	**p.Ala104Thr** (c.310G>A) *		9.6 ± 3.8	Function impaired	0.9153	pathogenic
	**p.Ala451Gly** (c.1352C>G) *****		104.6 ± 11.8	WT-like	0.1774	benign
	**p.Ala451Ser** (c.1351G>T) *****		96.1 ± 11.6	WT-like	0.2429	benign
	**p.Ala451Leu** (c.1351_2GC>CT) *		1.4 ± 2.5	Function impaired	0.9707	pathogenic

Bold font was used to distinguish between amino acid changes in the variants and the nucleotide change in the parentheses.

## Data Availability

The original contributions presented in the study are included in the article/Appendix A, further inquiries can be directed to the corresponding author.

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
