# Peer review of "Functional Studies of Deafness-Associated Pendrin and Prestin Variants"

_ijms, 2024, doi:10.3390/ijms25052759_

Round 1

Reviewer 1 Report

Comments and Suggestions for Authors

Review

to manuscript “Functional studies of deafness-associated pendrin and prestin 2 variants” by authors Satoe Takahashi, Takashi Kojima, Koichiro Wasano, and Kazuaki Homma

The authors presented the interesting work, which is a logical continuation of previous in vitro studies concerning the assessment of the pathogenetic influence of missense variants of pendrin and prestin to determine their association with hearing loss. This study focused on the functional impact of pendrin and prestin variants on the transport function of anions localized in the part of the core domain (1-4 and 8-11 segments) – in the transmembrane region of TM-9 and TM-10.

Major comment

1. Since, the core domain is not fully investigated (for example the TM-3 and TM-2 regions), I think, the authors must be pay a more attention to formulate the actual reasons of the study only in this part of the core domain (TM-9 and TM-10) of the pendrin and prestin proteins in the introduction section.

Minor comments

1. The abstract on the manuscript is contains many common phrases and does not reflect the details of the presented study. Please reform the abstract as described in the instructions for authors or represent briefly all part of the manuscript (background, aim of the study, material and methods, results, discussion and conclusion).

2. Keywords. The names of human genes should be written in capital letters, in italics. Please, check throughout by the text.

3. The conclusion section is missing. Please add the conclusion.

Recommendation

Accept after major comments.

Author Response

Reviewer #1

The authors presented the interesting work, which is a logical continuation of previous in vitro studies concerning the assessment of the pathogenetic influence of missense variants of pendrin and prestin to determine their association with hearing loss. This study focused on the functional impact of pendrin and prestin variants on the transport function of anions localized in the part of the core domain (1-4 and 8-11 segments) – in the transmembrane region of TM-9 and TM-10.

Major comment

  1. Since, the core domain is not fully investigated (for example the TM-3 and TM-2 regions), I think, the authors must be pay a more attention to formulate the actual reasons of the study only in this part of the core domain (TM-9 and TM-10) of the pendrin and prestin proteins in the introduction section.

Based on the recent structural studies, it is understood that TM10 within the core domain includes the anion binding site, which is crucial for the function of pendrin. TM9 connects to TM10, and many deafness-associated variants were found clustered in the region, implying high likelihood of these variants to be pathogenic. Since the objective of our experimental effort is to identify truly pathogenic variants that are causally associated with hearing loss, we prioritized our effort to characterize variants found in these regions in this study. We make this clear under the introduction section in our revised manuscript (highlighted in yellow).

Minor comments

  1. The abstract on the manuscript is contains many common phrases and does not reflect the details of the presented study. Please reform the abstract as described in the instructions for authors or represent briefly all part of the manuscript (background, aim of the study, material and methods, results, discussion and conclusion).

We have edited the abstract accordingly. Changes are highlighted in yellow in our revised manuscript.

  1. Keywords. The names of human genes should be written in capital letters, in italics. Please, check throughout by the text.

In the keywords, our intention was to refer to the protein names (non-italic capital letters), not genes. When we refer to human genes in the text, we made sure to use italics.

  1. The conclusion section is missing. Please add the conclusion.

We do not think that inclusion of a concluding statement greatly enhances the readability of this mostly descriptive functional study. Please note that the instructions on the template Word file (provided by the journal) stated that “This section (Conclusion) is not mandatory”.

Thank you for thoroughly reviewing our paper.

Reviewer 2 Report

Comments and Suggestions for Authors

I consider that the article is very well written and presented, that there is a lot of work behind it, and that the pathogenicity of the mutations is well demonstrated with the functional assays. As positions of cDNA and protein are mentioned in the article, I consider it necessary to add the reference NM numbers used for SLC26A4 and SLC26A5. In fact, in the case of prestin, it is crucial because there are different isoforms. Example: SLC26A4 (human) NM_000441.2

Author Response

Reviewer #2

I consider that the article is very well written and presented, that there is a lot of work behind it, and that the pathogenicity of the mutations is well demonstrated with the functional assays. As positions of cDNA and protein are mentioned in the article, I consider it necessary to add the reference NM numbers used for SLC26A4 and SLC26A5. In fact, in the case of prestin, it is crucial because there are different isoforms. Example: SLC26A4 (human) NM_000441.2

Thank you for your comments. The NM numbers used in this study was added to Materials and Methods section (highlighted in yellow).

Reviewer 3 Report

Comments and Suggestions for Authors

In this study, Takahashi et al. investigate the pathogenic functions of deafness-associated variants of pendrin and prestin. They utilize highly reproducible fluorometric antiport assays and nonlinear capacitance 59 (NLC) measurements in HEK293T cell lines, respectively. The authors explore 25 pendrin missense variants through doxycycline-induced antiport assays and reveal that many of these variants exhibit significantly reduced antiport activities. This finding suggests the importance of conserved residues in transmembrane domains 9 and 10 of the core domain for anion transporter function. Furthermore, the authors characterize three prestin variants associated with hearing loss (p.A100T, p.P119S, and p.S441L), using NLC as a proxy for their motility function. They discover that these mutations impair motor function. Interestingly, mutations in A100 in pendrin, equivalent to p.A100T in prestin, alter the anion transport function in pendrin, indicating the shared significance of these conserved residues for both pendrin and prestin.

The authors also verify these assays as relevant for assessing the pathogenicity of mutations, demonstrating consistency with results from a machine-learning-driven computational method, AlphaMissense (AM). Moreover, they identify several variants that were misclassified by AM as benign and argue that limitations in AM predictions, mainly due to its dataset relying heavily on structural information, are significant.

Although the authors utilize a heterologous expression system, this study provides invaluable insights into the mutations of prestin and pendrin. The study is well-designed, and the data quality is excellent. This study possesses the requisite quality for acceptance.

Author Response

Reviewer #3

In this study, Takahashi et al. investigate the pathogenic functions of deafness-associated variants of pendrin and prestin. They utilize highly reproducible fluorometric antiport assays and nonlinear capacitance 59 (NLC) measurements in HEK293T cell lines, respectively. The authors explore 25 pendrin missense variants through doxycycline-induced antiport assays and reveal that many of these variants exhibit significantly reduced antiport activities. This finding suggests the importance of conserved residues in transmembrane domains 9 and 10 of the core domain for anion transporter function. Furthermore, the authors characterize three prestin variants associated with hearing loss (p.A100T, p.P119S, and p.S441L), using NLC as a proxy for their motility function. They discover that these mutations impair motor function. Interestingly, mutations in A100 in pendrin, equivalent to p.A100T in prestin, alter the anion transport function in pendrin, indicating the shared significance of these conserved residues for both pendrin and prestin.

The authors also verify these assays as relevant for assessing the pathogenicity of mutations, demonstrating consistency with results from a machine-learning-driven computational method, AlphaMissense (AM). Moreover, they identify several variants that were misclassified by AM as benign and argue that limitations in AM predictions, mainly due to its dataset relying heavily on structural information, are significant.

Although the authors utilize a heterologous expression system, this study provides invaluable insights into the mutations of prestin and pendrin. The study is well-designed, and the data quality is excellent. This study possesses the requisite quality for acceptance.

Thank you for your comments.

Round 2

Reviewer 1 Report

Comments and Suggestions for Authors

No comments